# Functional Characterization of Muscarinic Receptors in Human Schwann Cells

**DOI:** 10.3390/ijms21186666

**Published:** 2020-09-11

**Authors:** Roberta Piovesana, Alessandro Faroni, Ada Maria Tata, Adam J. Reid

**Affiliations:** 1Department of Biology and Biotechnology “C. Darwin”, Sapienza University of Rome, P.le Aldo Moro, 5, 00185 Rome, Italy; roberta.piovesana@uniroma1.it; 2Blond McIndoe Laboratories, Division of Cell Matrix Biology and Regenerative Medicine, School of Biological Sciences, Faculty of Biology, Medicine and Health, The University of Manchester, Manchester Academic Health Science Centre, Manchester M13 9PT, UK; alessandro.faroni@manchester.ac.uk; 3Research Centre of Neurobiology “Daniel Bovet”, Sapienza University of Rome, P.le Aldo Moro, 5, 00185 Rome, Italy; 4Department of Plastic Surgery & Burns, Wythenshawe Hospital, Manchester University NHS Foundation Trust, Manchester Academic Health Science Centre, Manchester M23 9LT, UK

**Keywords:** human Schwann cells, acetylcholine, muscarinic receptors, differentiation, glial cells

## Abstract

Functional characterization of muscarinic cholinergic receptors in myelinating glial cells has been well described both in central and peripheral nervous system. Rat Schwann cells (SCs) express different muscarinic receptor subtypes with the prevalence of the M2 subtype. The selective stimulation of this receptor subtype inhibits SC proliferation, improving their differentiation towards myelinating phenotype. In this work, we describe for the first time that human SCs are cholinoceptive as they express several muscarinic receptor subtypes and, as for rat SCs, M2 receptor is one of the most abundant. Human SCs, isolated from adult nerves, were cultured in vitro and stimulated with M2 muscarinic agonist arecaidine propargyl ester (APE). Similarly to that observed in rat, M2 receptor activation causes a decreased cell proliferation and promotes SC differentiation as suggested by increased Egr2 expression with an improved spindle-like shape cell morphology. Conversely, the non-selective stimulation of muscarinic receptors appears to promote cell proliferation with a reduction of SC average cell diameter. The data obtained demonstrate that human SCs are cholinoceptive and that human cultured SCs may represent an interesting tool to understand their physiology and increase the knowledge on how the cholinergic stimulation may contribute to address human SC development in normal and pathological conditions.

## 1. Introduction

During development, the close relationship between neurons and Schwann cells (SCs), the main glial cell population in peripheral nervous system (PNS), is mediated by different molecules including acetylcholine (ACh) [1,2,3,4,5]. In particular, in nerve fibres of invertebrates, ACh receptors on SCs localize to the axon-SCs boundary [6]. Myelinating glia cells express both muscarinic and nicotinic ACh receptor subtypes during development and in adulthood [1]. Several studies demonstrated that ACh signalling, via muscarinic receptor activation, is able to modulate proliferation and differentiation in astrocytes [7] and oligodendrocytes [8].

Cultures of SCs, obtained from sciatic nerve of 2 days post-natal rats express muscarinic receptors (M1–M4) [9]. In particular, M2 mAChR subtype is the most abundant and persists in mature SCs [10]. 

SC treatment with cholinergic agonists, triggers significant changes in intracellular inositol 1,4,5-triphosphate (IP3) and cyclic adenosine monophosphate (cAMP) levels [9]. Cells treated with a selective agonist of M2 muscarinic subtype arecaidine propargyl ester (APE) showed an arrest of SC proliferation in the G1-S phase [11] supported by the negative modulation of c-Jun and Notch-1, thereby reducing SC proliferation [10]. M2 receptor stimulation promotes SC differentiation towards myelinating phenotype with the upregulation of promyelinating transcription factors as Sox10 and Egr2, followed by an increased expression of myelin proteins (e.g., myelin basic protein, MBP; myelin protein zero, P0 and peripheral myelin protein 22, PMP22) [10]. Ultrastructural analyses by transmission electron microscopy (TEM) of the sciatic nerves of M2/M4 knock-out mice have demonstrated the presence of degenerating axons and alterations in myelin organization of the medium/large axons [10]. Moreover, cholinergic stimulation, but most evidently the activation of M2 receptor, modulates nerve growth factor (NGF) levels, negatively regulating the proNGF-B isoform involved in apoptosis [12].

These results clearly demonstrate that, at least in the rat model, M2 mAChRs negatively modulate proliferation and upregulate promyelinating transcriptional factors and myelin protein expression, promoting SC differentiation.

It is relevant to note that human SCs (hSCs) have been poorly investigated. Increasing knowledge on hSCs in this field is of relevance, considering the role of SCs in different peripheral pathologies including neurofibromatosis and peripheral neuropathies [13,14,15]. Starting from the described role of M2 receptor in rat SC physiology, in the present study we have analysed cultures of hSCs derived from sensory nerves withdrawn from human adipose tissue. As observed in the rat model, cholinergic receptors are expressed also in hSCs. The preferential activation of M2 receptor negatively modulates hSC proliferation without affecting cell survival and supporting cell differentiation. Instead, the non-selective muscarinic receptors stimulation appears to promote cell proliferation with a reduction of the average cell diameter. These data highlight that hSCs, similarly to what observed in rat SCs, are cholinoceptive and cholinergic stimulation may be involved in hSC physiology without affecting their plasticity. 

## 2. Results

### 2.1. Muscarinic Receptors Are Expressed in hSCs 

Human SCs, obtained from 3 different patients, were first characterized for the expression of S100β, as SC marker. As showed in Figure 1A, the cells presented a more pronounced spindle-like morphology, and the 90% of the cells in cultures resulted immune-positives for S100β. 

Then, the expression of cholinergic muscarinic receptor subtypes was evaluated by RT-PCR analysis. As reported in Figure 1B, in hSCs from 3 different patients, M1, M2, and M3 subtypes were expressed at higher levels, whereas M4 and M5 expression appeared to be variable between different patients. Similar results were obtained by qRT-PCR analysis (Appendix A). M2 subtype transcripts were present in all patients, and its expression at protein level has been confirmed by Western blot analysis. As shown in Figure 1C, M2 muscarinic subtype was expressed in all patients albeit at variable levels and with evident glycosylation pattern of the receptors between different patients. Cell cultures obtained from these three patients were stimulated for 3 days in vitro with M2 agonist APE; this agonist has been largely characterized in different murine and human cell lines where its ability to specifically bind M2 receptor subtype was largely demonstrated [10,16,17]. As reported in Figure 1D, M2 stimulation with 100 µM APE resulted in a significant decrease of cell growth in all patients after 3 days of treatment. 

### 2.2. Analysis of Cell Growth, Survival, and Morphology

In order to evaluate the ability of muscarinic receptors to modulate hSC development, we analysed the cell growth by MTS assay for up to 7 days of 100 µM APE or muscarine treatments in more patients (*n* = 5) (Figure 2A). APE treatment decreased cell growth after 72 h of treatment, remaining substantially lower if compared with untreated cells at 7 days of treatment. Instead, the non-selective agonist muscarine, used at the same final concentration of 100 µM, promoted cell growth after 5 days of treatment in vitro (DIV), albeit an initial decrease of cell number after 3 days of treatment was evident (Figure 2A). Statistical analysis between different time points, reported in the Appendix A, showed that although a significant increase of cell growth between different time points (e.g., APE 3 DIV vs. APE 7 DIV) was observed, cell growth decreased between APE treatment and untreated cells at every time point (Appendix A). Considering this apparent increase of cell growth upon 7 days of 100 µM APE treatment, in order to evaluate if the effect may be dependent on reduced activity of M2 agonist during 7 days in vitro, we performed the same experiment at different concentrations of APE, modifying the experimental plan with the media change with or without APE treatment at the third day of treatment. In this experimental condition, differently to what was observed in the previous experiment reported in Figure 2A, we observed that the cell growth was unchanged at 3 DIV and 7 DIV after 100 µM APE treatment, confirming the inhibitory effect of the high dose of APE on cell growth. Moreover, the results obtained indicated that only APE at concentrations of 50 and 100 µM was able to significantly reduce the cell growth but the 50 µM APE effect was evident only after 7 days of treatment, whereas lower concentrations (25 μM) did not show any effects (Figure 2B). Similarly, the analysis of cell growth at different concentrations of muscarine (ranging from 25 to 100 µM) demonstrated that the low doses of the non-selective agonist did not show any effects on cell growth (data unpublished) and that only the concentration of 100 µM was able to positively modulate cell proliferation (Figure 2A).

The cholinergic treatments were not toxic for hSCs; trypan blue staining was used to determine possible drug toxicity; both agonists did not show toxic effects on hSC vitality at 24 h of treatment (Figure 3A–D). Morphological analysis, performed using Cell Countess, showed a significant increase of the average cell diameter after M2 stimulation (Figure 3F,H). Conversely, muscarine showed an opposite effect, reducing the average cell diameter of the cells (Figure 3G,H). Moreover, APE treatment caused a lower density of cells compared to the untreated cells (Figure 3I,J), supporting the reduction of cell growth observed by MTS assay. Interestingly, hSCs treated with APE also showed a more elongated morphology (Figure 3J). This result was supported by a significant upregulation of Egr2 protein levels, a typical promyelinating transcriptional factor, after M2 agonist exposure (Figure 3L,M).

## 3. Discussion

During development and peripheral nerve injury, SCs produce and secrete a variety of neurotrophic factors and extracellular matrix components that provide a helpful microenvironment for neuronal survival and axon elongation [18]. Transplanted SCs can improve nerve regeneration and then myelinate the regenerated axons, although their isolation is difficult and requires healthy nerve sacrifice. 

Our knowledge on SC development and plasticity is derived mainly from rodent models (rat and mice), whereas hSCs have been poorly investigated. Understanding hSC physiology could be relevant for the improvement of therapeutic interventions in peripheral pathologies such as neuropathies and in peripheral nerve injury (PNIs). 

In the rat model, CNS and PNS myelinating glia are cholinoceptive and express both muscarinic and nicotinic cholinergic receptors at different times of development [7,8,9,10,11]. Moreover, also after nerve injury, ACh plays a significant role in motor nerve terminal outgrowth and muscle repair. In fact, the inhibition of postsynaptic neuromuscular ACh receptors by α-bungarotoxin (α-Btx) in the skeletal muscles significantly reduces nerve outgrowth [19], suggesting that a local release of ACh is essential to directly trigger nerve terminal outgrowth [3]. 

Cultures of rat immature SCs express muscarinic receptor (M1–M4), with a higher expression of M2 mAChR subtype, which increases the expression of myelin proteins [9,10], addressing SCs towards promyelinating phenotype. ACh secreted by the nerve terminal can also modulate perisynaptic SCs (PSCs), which surround nerve terminals at the neuromuscular junction, and regulates glial fibrillary acidic protein (GFAP) expression and cytoskeletal changes via mAChRs [20]. 

In the present work, for the first time, we demonstrate that hSCs are cholinoceptive; in fact, they express several muscarinic receptor subtypes, albeit the expression levels of the single subtypes are variable between different patients. Similarly to that observed in rat, they express M2 muscarinic receptor that is also differentially expressed in different patients. However, M2 agonist, APE, is able to decrease SC growth without affecting cell viability, as indicated by trypan blue staining. In particular, 100 µM APE is able to arrest cell growth during 7 DIV. Moreover, we have observed that 50 µM APE is also able to decrease cell growth only after 7 DIV. This may be dependent on distinct reasons: 1. the affinity of the M2 receptor for APE may vary according to the glycosylation grade of the receptors in the different patients, as observed in Western blot analysis and 2. the prolonged exposure to 50 µM APE may be necessary to activate the M2 downstream signalling transduction pathway. 

According to their physiological role, SCs change their morphology, being much longer when they are induced towards a myelinating phenotype [21]. Morphological analyses show that hSCs cultured in growth condition, in presence of forskolin (Fsk) and glial growth factor 2 (GGF-2) in the culture media, have a classic bipolar morphology. According to our previous data gathered in the rat model [10,11], APE treatment elongates hSCs with a significant increase of the average cell diameter and Egr2 protein expression, suggesting that the M2 selective stimulation enhances promyelinating phenotype also in hSCs [10]. The ability of muscarinic receptors to modulate Egr2 transcription factor family is well documented [22,23]. Interestingly the modulation of Egr2 by M2 receptors may at least explain the inhibitory effect on cell proliferation. SC proliferation is in fact supported by Neuregulin 1 (NRG1) pathway and c-jun expression. Our previous studies in rat have largely demonstrated the ability of M2 receptors to downregulate cyclic adenosine monophosphate (cAMP) levels and c-jun expression [9,17]. It is possible to hypothesize that the same mechanism was maintained in hSCs. Moreover c-jun and Egr2 have a cross-antagonistic relationship [24], thus the upregulation of Egr2-M2 mediated, may indirectly contribute to a decrease of c-jun expression and subsequent SC proliferation. 

In the present study, it has not been possible to perform pharmacological competition studies with selective antagonists for muscarinic receptors considering the limited number of the cells obtained from the same patients. Albeit the APE-derived compounds were reported as preferential M1–M4 agonists [25], several structural variations of the molecules appeared to improve the selectivity for M2 receptor [26,27]. In particular, the M2 agonist APE has been largely characterized by our group by pharmacological competition studies performed in rat glial cells and DRG sensory neurons, clearly demonstrating that APE effects are largely counteracted only by gallamine, one of the preferential M2 receptor antagonist [8,28]. Moreover, in neuroblastoma and glioblastoma cell lines, we have also demonstrated that M1 and M3 antagonists, pirenzepine and 4-DAMP, respectively, were not able to counteract APE effects, confirming preferential selectivity of APE for M2 receptor. Moreover, after M2 knockdown by short interference RNA, APE effect was completely abolished [16,29,30,31]. Finally, it is relevant to note that M4 receptor expression both in rat [9] and in hSCs appears significantly lower than M2 subtype, at least at transcript level; for this reason, the potential interference of M4 receptors, should be irrelevant.

Muscarine treatment, instead, promotes cell proliferation, whereas the cell morphology after non-selective agonist treatment, remains similar to untreated cells, albeit the reduction of cell diameter observed. The contrasting results obtained with muscarine are likely dependent on the ability of muscarine to bind all muscarinic receptor subtypes. The positive effects of muscarine on cell proliferation may be dependent on the simultaneous activation of Gq11-proteins, probably M1- and M3-mediated ones. The reduction of the cell size may be dependent on more intense SC proliferation after muscarine treatment. In fact, when the cells are in proliferation, they assume a rounded morphology, and after the mitotic cell division, they have a reduced dimension. 

## 4. Materials and Methods 

### 4.1. Human Schwann Cells Cultures

Human nerves were isolated from adipose tissue of patients undergoing reconstructive surgery at Wythenshawe Hospital, Manchester University NHS Foundation Trust, UK, after informed consent was granted. All patients were female, healthy, and aged 44–64 years. All procedures were approved by the National Research Ethics Committee, UK (NRES 13/SC/0499) and conformed with the World Medical Association Declaration of Helsinki. 

Human nerves were dissected and immersed into high-glucose DMEM (Sigma-Aldrich, St. Louis, MO, USA) containing 10% (*v*/*v*) foetal bovine serum (FBS, LabTech, Uckfield, UK), 2 mM l-glutamine (GE Healthcare, Chicago, IL, USA), and 1% (*v*/*v*) penicillin–streptomycin solution (SC media). Connective tissue was removed with tweezers and the single fibres extracted were cut in small pieces (approximately 1 mm each) and cultured in a 60 cm^2^ dish with SC media supplemented with 10 µM forskolin (Fsk; Sigma-Aldrich) + 100 ng/mL glial growth factor 2 (GGF-2, Acorda Therapeutics, Ardsley, NY, USA) for 2 weeks. Media were changed every 3 days. After 2 weeks of culture, human nerves were digested with Dispase (Life Technologies, Carlsbad, CA, USA)/Collagenase IV (Life Technologies), in a ratio of 1:1, for 24 h. The day after the solution containing nerves partially digested was collected in 15 mL falcon tube and nerves were gently triturated using a glass pipette. Cell suspension was passed through a sterile 70 µm mesh to remove clumps of undissociated tissue and axon debris and then centrifuged at 900 rpm for 5 min. Cell pellet was gently resuspended in SC media supplemented with 10 µM forskolin (Fsk, Sigma-Aldrich) + 100 ng/mL glial growth factor 2 (GGF-2, Acorda Therapeutics, Ardsley, NY, USA) and plated in a T75 flask. T75 flask was previously coating with Poly-d-Lysine (Sigma-Aldrich, P7280) for 30 min at RT and subsequently coating with laminin (10 mg/mL, Sigma-Aldrich) for 2–3 h at 37 °C. Cells were maintained in T75 flasks at 37 °C and 5% CO_2_, with three medium changes every week. 

### 4.2. Pharmacological Treatments 

Human SCs were treated with M2 selective agonist arecaidine propargyl ester hydrobromide (APE, A140; Sigma-Aldrich) at the final concentration ranging from 25 to 100 µM. Albeit APE was reported as preferential M1–M4 agonist [25], several modifications of the molecule suggested a preferential agonistic effect on M2 receptor subtype [26,27]. Our works have been focused on the characterization of APE hydrobromide by pharmacological competition studies with M2 antagonists methoctramine or gallamine [8,17,32]. Moreover, the silencing of M2 receptors by siRNAs have largely demonstrated the selectivity of APE for M2 receptor in several cell lines (i.e., human glioblastoma, neuroblastoma, and urothelial bladder cell lines [16,29,30,31,33]. 

To study all muscarinic receptors activation, hSCs were treated with the non-selective muscarinic agonist, muscarine (Sigma-Aldrich), at the final concentration ranging from 25 to 100 µM. 

All experiments were performed in technical and experimental triplicate. 

### 4.3. Cell Proliferation and Survival Assay

Human SCs from different patients were plated in 24-well plates (Corning Life Science, Amsterdam, Netherlands) at a density of 20 × 10^3^ cells/well. The day after plating, cells were treated with both muscarinic agonists, APE and muscarine, as mentioned above. Cell proliferation was evaluated at 3, 5, and 7 days of treatment. Cell medium was aspirated, and cells were incubated in 20% (*v*/*v*) CellTiter 96 AQueous One Solution Cell Proliferation Assay, [3-(4,5-dimethylthiazol-2-yl)-5-(3-carboxymethoxyphenyl)-2-(4-sulfophenyl)-2H-tetrazolium, inner salt; MTS; Promega, Southampton, UK ), diluted in phenol-free DMEM (Sigma-Aldrich). Following 2 h of incubation at 37 °C, the absorbance at 490 nm was recorded with an Asys UVM-340 microplate reader/spectrophotometer (Biochrom, Cambridge, UK). After the standard curve was performed, data were expressed as cell number ± SEM. 

Under the same experimental conditions, we evaluated the presence of dead cells. After trypsinization, the percentage of dead cells was read with Cell Countess (Thermo Fisher Scientific, Waltham, MA, USA), using trypan blue staining (1:10 *v*/*v*, Thermo Fisher Scientific, Altrincham, UK). Cell Countess analysed live (green) and dead (red) cells (as shown in Figure 3A–C); moreover, Cell Countess reported every analysed diameter in a graph (Figure 3E–G) in which the green columns represent the live cell diameter and the red columns represent the dead cell diameter. The results were reported in a column graph, and ANOVA analysis followed by Tukey post hoc test was performed.

### 4.4. RNA Extraction and RT-PCR Analysis

Cells were collected and stored in RNA cell protect agent (Qiagen, Hilden, Germany). Total RNA was isolated from hSCs using RNeasy Plus Mini Kit (Qiagen), according to the manufacturer’s protocol. RNA was quantified using a NanoDrop ND-100 spectrophotometer (Thermo Fisher Scientific). Each sample was reverse-transcribed using RT2 First Strand Kit (Qiagen), according to the manufacturer’s instructions. RT-PCR was performed with GoTaq Green Master Mix (Promega) using MultiGene Optimax (Labnet, Edison, NJ, USA). The sequences of the used primers are reported in Table 1. Moreover, 18s was used as housekeeping gene. 

### 4.5. Protein Extraction and Western Blot

Whole-cells lysates were obtained by scraping cells from confluent 6 well-plates using RIPA Buffer (Sigma-Aldrich) supplemented with protease and phosphatase inhibitors (Thermo Scientific). Lysates were incubated for 30 min on ice and later centrifuged for 20 min at 14,000 rpm at 4 °C. According to the manufacturer’s protocol, protein concentrations were determined using Pierce™ BCA Protein Assay Kit (Thermo Fisher Scientific). Sample buffer (6×) was added to protein lysates and heated for 5 min at 100 °C. 

Precisely, 30 μg of each sample was loaded onto a 10% SDS (Sodium dodecyl sulphate) polyacrylamide gel and run at 120 V using running buffer (25 mM Tris, 190 mM glycine, 0.08% (*w*/*v*) SDS). SDS-PAGE gels were transferred for 60 min onto nitrocellulose blotting membranes (GE Healthcare Life Science) at 80 V in transfer buffer (25 mM Tris-base; 192 nM glycine, 20% (*v*/*v*) methanol). After transfer, membranes were blocked for 1 h in blocking buffer (Tris-buffer saline (TBS)-Tween solution containing 5% non-fat dry milk). Membranes were incubated with the primary antibody, diluted in blocking buffer, overnight at 4 °C.

Primary antibodies used were: mouse anti-M2 antibody (1:500, Abcam, Cambridge, UK) and rabbit anti-Egr2 (1:500, ProteinTech, Manchester, UK). After overnight incubation, membranes were washed 5 times with TBS-Tween buffer and incubated for 1 h at RT with secondary antibody: anti-rabbit horseradish peroxidase (1:2000, Cell Signaling, Hitchin, UK) or anti-mouse horseradish peroxidase (1:1000, Cell Signaling, Hitchin, UK) for chemiluminescence detection. To determine housekeeping protein, membranes were stripped before reblotting with another primary antibody. β-tubulin was used as protein reference (rabbit anti-β-tubulin, 1:1000, Abcam, Cambridge, UK). 

Membranes were exposed to SuperSignal West Pico Chemiluminescent Substrate (Thermo Fisher Scientific) for signal detection. The bands were detected by exposition to ChemiDoc (Molecular Imager ChemiDoc XRS + System with Image Lab Software, Bio-Rad, CA, USA). The optical density (OD) of each protein band was analysed with ImageJ software (National Institutes of Health, NIH, 469 Bethesda, MD, USA) and normalized against the OD of the protein reference band. 

### 4.6. Immunocytochemistry Analysis

After fixation with 4% paraformaldehyde (PFA, Sigma-Aldrich), cells were incubated with Triton X-100 0.2% for 30 min at RT. Then, cells were washed twice with PBS and treated for 1 h at RT with block solution containing PBS + 0.1% Triton X-100 and, according with secondary antibody, 10% normal donkey serum (NDS). Cells were incubated with primary antibody (polyclonal anti-S100β, Dako, Glostrup, Denmark) in 0.1% Triton X-100, 0.1% (*w*/*v*) BSA, and 0.1% (*w*/*v*) sodium azide in PBS at 4 °C overnight. The day after, cells were washed with PBS three times for 10 min and incubated with secondary antibody (Donkey anti-rabbit IgG (H + L) Highly Cross-Adsorbed Secondary Antibody, Alexa Fluor 568, Life Technologies) in 0.1% Triton X-100, 0.1% (*w*/*v*) BSA, and 0.1% (*w*/*v*) sodium azide in PBS for 1 h at RT. Then, cells were washed 3 times with PBS and mounted with Vectashield mounting medium for fluorescence containing 4′-6′-diamidino-2-phenylindole for nuclear staining (H1200, Vector Lab, DBA, Milan, Italy). Images were taken using a fluorescence microscope (Olympus IX51, Southend-on-Sea, UK).

### 4.7. Data Analysis 

Data analyses were performed with GraphPad Prism 8 (GraphPad Software, La Jolla, CA, USA). Data were presented as the mean ± standard error of the mean (SEM). Student’s *t*-test or one-way ANOVA analyses followed by Tukey or Dunnett post hoc tests were used. A value of *p* < 0.05 was considered statistically significant: * *p* < 0.05, ** *p* < 0.01, *** *p* < 0.001, and **** *p* < 0.0001. The densitometric analysis of Western blot was performed by ImageJ software (National Institutes of Health, NIH, 469 Bethesda, MD, USA).

## 5. Conclusions

The data obtained in the present work represent the first evidence that hSCs can be isolated by peripheral nerves present in human subcutaneous adipose tissue. Moreover, we demonstrate that they are cholinoceptive and that ACh, via muscarinic receptors, may differently contribute to axon-SCs cross-talk also in human. These results may be of great relevance in future studies to design new therapeutic intervention for the treatment of traumatic peripheral nerve injuries or for peripheral nervous system pathologies such as neurofibromatosis and the peripheral neuropathies.

## Figures and Tables

**Figure 1 ijms-21-06666-f001:**
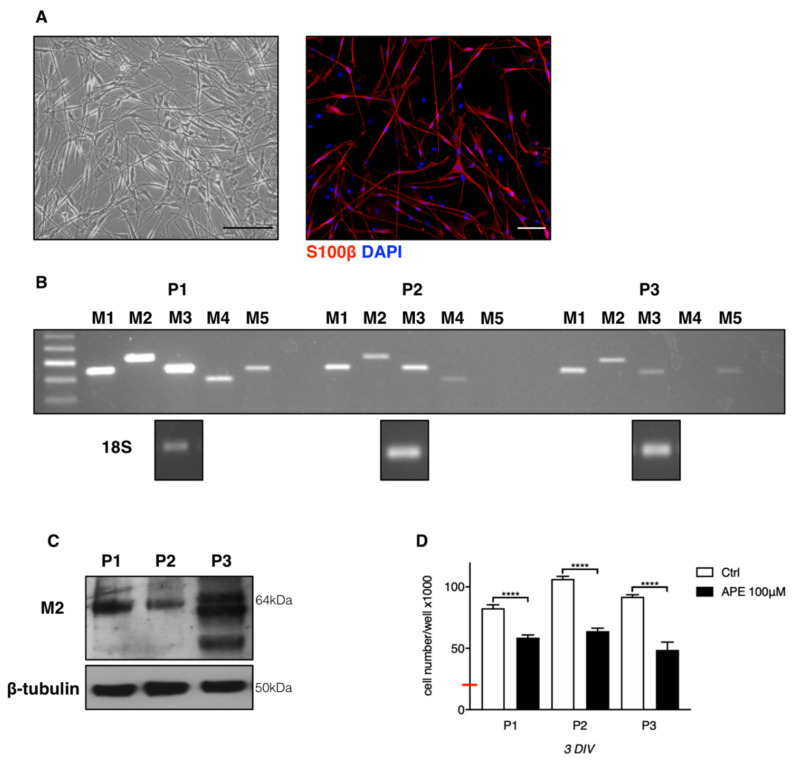
(**A**) Human Schwann cells (hSCs) were photographed at contrast-phase microscope (scale bar = 100 µm); hSCs show a typical bipolar morphology. Immunostaining for S100β appeared present in the 90% of the cultured cells (scale bar = 100 µm). (**B**) Analysis by RT-PCR of muscarinic receptors subtypes in hSCs obtained from 3 different patients (P1, P2, and P3). Here, 18s was used as housekeeping gene. (**C**) M2 muscarinic receptor expression of the protein samples obtained from the same patients analysed by RT-PCR. β-tubulin was used as reference protein. (**D**) MTS assay showing the ability of M2 agonist APE to reduce cell growth in all three patients. The results are the average ± SEM of three independent experiments performed in triplicate (**** *p* < 0.0001). The red bar represents the number of cells plated (20 × 10^3^ cells/well). The day after plating, cells were treated with 100 µM APE and MTS assay was performed after 3 days of treatment.

**Figure 2 ijms-21-06666-f002:**
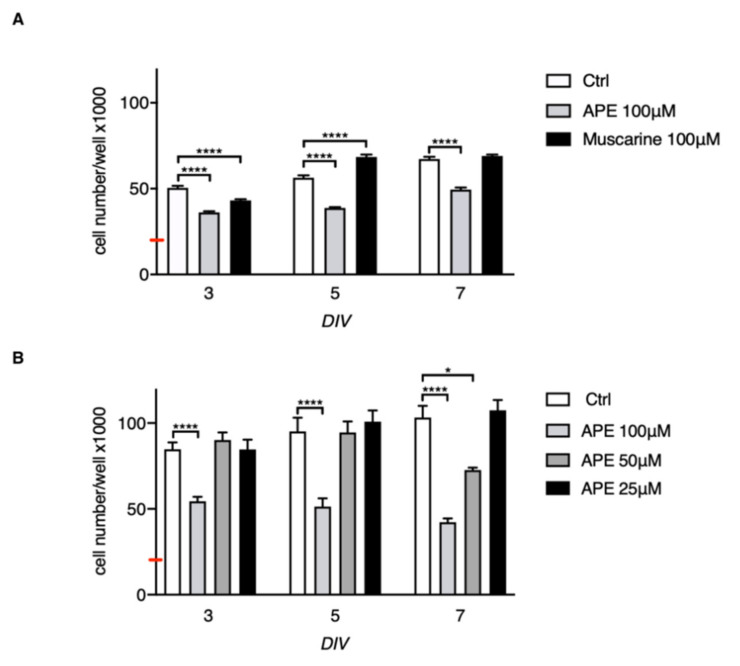
MTS assay in hSCs maintained up to 7 days of treatment in vitro (DIV) in presence or absence of muscarinic agonists. (**A**) APE (100 µM) exposure was able to decrease cell growth up to 7 days of treatment (**** *p* < 0.0001; APE vs. ctrl at the same time point). The treatment with 100 µM muscarine appeared to promote cell proliferation after 5 days of treatment (**** *p* < 0.0001; muscarine vs. ctrl at the same time point). Data are represented as mean ± SEM of three independent experiments performed in triplicate in five different patients. One-way ANOVA with Tukey’s post hoc test between different time points is reported in the Appendix A. (**B**) hSCs cell growth was also evaluated at different APE concentrations (25, 50, and 100 µM) at 3, 5, and 7 DIV. ANOVA with Dunnett multiple comparison analyses was performed against control at the same time point. Data are represented as mean ± SEM of three independent experiments performed in triplicate in five different patients (**** *p* < 0.0001; * *p* < 0.05). The red bar represents the number of cells plated (20 × 10^3^ cells/well). The day after plating, cells were treated with both muscarinic agonists at the final concentration of 100 µM (**A**) or with different concentrations of APE (**B**); MTS assays were performed after 3, 5, and 7 days of treatment.

**Figure 3 ijms-21-06666-f003:**
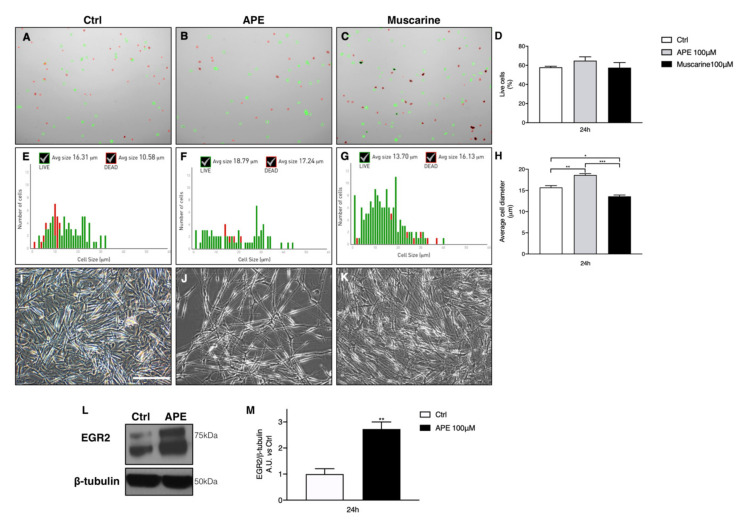
Analysis of cell death after trypan blue assay, using Cell Countess. (**A**–**D**) The analysis showed that both cholinergic agonists are not toxic for hSCs ((**A**–**C**), green dots are live cells, whereas red dots are dead cells). In (**D**), the percentage of the live cells under different experimental conditions is reported. (**E**–**G**) Countess analysis shows a significant increase of average cell diameter. The graphs show the analysis of cell diameter, where the green columns and the red columns represent the live cells or dead cells, respectively, presenting a specific diameter. Up the panels, the analyses show the average cell diameter for live and dead cells ((**E**) control; (**F**) APE; (**G**) muscarine)). In (**H**), the average of cell diameter is reported; APE induces an increase of cell diameter after 24 h of treatment (** *p* < 0.01 APE vs. ctrl; APE = 18.63 ± 0.33 µm; *** *p* < 0.01 APE vs. Muscarine), whereas muscarine exposure induces a reduction in average cell diameter (* *p* < 0.05 muscarine vs. ctrl; muscarine: 13.60 ± 0.32 µm). (**I**–**K**) Images in phase contrast showing hSC morphology and density after APE and muscarine treatments (scale bar: 100 µm); ((**I**) control; (**J**) APE; (**K**) muscarine)). (**L**,**M**) The analysis of Egr2 protein expression, by Western blot analysis, has shown that APE induces a significant upregulation of Egr2 protein expression (** *p* < 0.01, APE treatment vs. ctrl = 2.73 ± 0.27). β-tubulin was used as protein reference. Data are represented as mean ± SEM of three independent experiments performed in three different patients.

**Table 1 ijms-21-06666-t001:** List of RT-PCR primers.

Gene	Forward 5′–3′	Reverse 5′–3′
*RN18S1*	ATCGGGGATTGCAATTATTC	CTCACTAAACCATCCAATCG
*M1*	CAGCAGTACCGAACCACGTA	CTCCTGACTTCCTGCCTAAA
*M2*	CCAAGACCCCGTTTCTCCAAG	CCTTCTCCTCTCCCTGAACAC
*M3*	CGCTCCAACAGGAGGAAGTA	GGAGTTGAGGATGGTGCTGT
*M4*	AATGAAGCAGAGCGTCAAGAA	TCATTGGAAGTGTCCTTATCA
*M5*	CCTGGCTGATCTCCTTCATC	GTCCTTGGTTCGCTTCTCTG

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
