# Peer review of "Functional Characterization of Muscarinic Receptors in Human Schwann Cells"

_ijms, 2020, doi:10.3390/ijms21186666_

Round 1

Reviewer 1 Report

Authors attempted to postulate that the assay using cultured Schwann cells (SCs) is useful tool for investigation of human SC development and physiology in normal and pathological condition because in this manuscript they showed that M2 receptor activation in cultured human SCs caused an inhibition of cell proliferation and promoted SC differentiation. Their approaches, using cultured SCs, are interesting and important, but there are some issues to be addressed in this manuscript.

1) Authors used isolated human and rat SCs in this manuscript. The manuscript must contain precious data that isolated cells can express SCs’ characters, using antibodies for their markers.

2) Most readers would like to know why the M2 receptor activation resulted in the inhibition of SC proliferation. Explanation and discussion must be added.

3) Authors must re-write in the discussion section because interpretation of their data was insufficient.

Author Response

Replies to reviewer’s comments

The authors thank all the reviewers for their comments and suggestions.

We have modified our manuscript in all parts indicated in red in the revised version, expanded discussion, modified Fig. 1 and 3, and added supplementary Fig. 1 and some new references, where required.

Rev 1

Authors attempted to postulate that the assay using cultured Schwann cells (SCs) is useful tool for investigation of human SC development and physiology in normal and pathological condition because in this manuscript they showed that M2 receptor activation in cultured human SCs caused an inhibition of cell proliferation and promoted SC differentiation. Their approaches, using cultured SCs, are interesting and important, but there are some issues to be addressed in this manuscript.

1) Authors used isolated human SCs in this manuscript. The manuscript must contain precious data that isolated cells can express SCs’ characters, using antibodies for their markers.

Thank you for the suggestion. We have enclosed in Fig.1 immunostaining for S100β in cultured human Schwann cells. As observable in the fig. 1A, the 90% of cells present in the culture are positive for Schwann cell marker, S100β.

2) Most readers would like to know why the M2 receptor activation resulted in the inhibition of SC proliferation. Explanation and discussion must be added.

The discussion has been expanded. We have added an explanation of the possible mechanisms-M2 receptor mediated that could negatively modulate Schwann cell proliferation

3) Authors must re-write in the discussion section because interpretation of their data was insufficient.

The discussion has been modified in several parts

Reviewer 2 Report

The article “Functional characterisation of muscarinic receptors in human Schwann cells” by Piovesana describes the expression of muscarinic M1-M5 receptors in human Schwann cells, and claims that the stimulation of M2 inhibits cell growth over 3-7 days.

The interest of this communication is based on the human origin of the cells. The "functional" characterization of muscarinic receptors is only limited to an effect on cell growth, which calls into question the title of the article (which should instead be "M2 muscarinic receptors inhibit human Schwann cell growth").

Figure 1C is incomplete. Shown is the cell growth after 3 days. Please add the cell number at day 1 to show the difference between day 1 and day 3.

Why didn’t you use a specific antagonist of mAChRs to make sure the growth inhibitory effect of APE is a consequence of a muscarinic stimulation? Please justify. The same applies for Fig. 2.

I am quite surprised that the paradoxical results obtained with APE and muscarine are only very partially discussed (L181-183) and without any reference. This part needs to be both improved and better discussed. I ask the authors to rewrite it by arguing it. Moreover, the discussion does not in any way consider explaining the results obtained: how does the specific stimulation of M2 mAChRs lead to an inhibition of cell growth? Mention should be made of the intracellular signaling induced by the activation of M2. This point is not covered in the article. The level of the discussion is poor as it is.

Minor

L46: IP3 and cAMP levels

L53: have demonstrated

L55: “modulates NGF production and maturation”. Maturation of what ? Central nervous system ?

L77: As showed in Fig. 1B, M2

Author Response

Replies to reviewer’s comments

The authors thank all the reviewers for their comments and suggestions.

We have modified our manuscript in all parts indicated in red in the revised version, expanded discussion, modified Fig. 1 and 3, and added supplementary Fig. 1 and some new references, where required.

Rev. 2

The article “Functional characterisation of muscarinic receptors in human Schwann cells” by Piovesana describes the expression of muscarinic M1-M5 receptors in human Schwann cells, and claims that the stimulation of M2 inhibits cell growth over 3-7 days.

The interest of this communication is based on the human origin of the cells. The "functional" characterization of muscarinic receptors is only limited to an effect on cell growth, which calls into question the title of the article (which should instead be "M2 muscarinic receptors inhibit human Schwann cell growth").

Thank you for your comments. The focus of this first paper on human Schwann cells was to analyse the muscarinic receptor expression and the first characterization of the non-selective and M2 selective receptors activation. For both the agonists we have analysed the cell growth, the cell survival (live-dead assay) and the cell morphology (see Fig. 1B, 2A and fig. 3). Only Egr2 expression was evaluated after APE exposure.

For this reason we think that the title could be conserved.

Figure 1C is incomplete. Shown is the cell growth after 3 days. Please add the cell number at day 1 to show the difference between day 1 and day 3.

The number of the cells plated is reported in Material and methods section. However in Fig. 1C (new Fig.1D) we have added a red bar on y axis indicating the cell number plated. DIV indicates the days on vitro after treatment. The treatment was added 24h after cells plating.

Why didn’t you use a specific antagonist of mAChRs to make sure the growth inhibitory effect of APE is a consequence of a muscarinic stimulation? Please justify. The same applies for Fig. 2.

The ability of APE to selectively bind M2 receptors has been previously characterized in rat Schwann cells using M2 antagonists gallamine and methoctramine (see Loreti et al, 2007, Uggenti et al, 2014 and Piovesana et al, 2019). Moreover we have also demonstrated that the antagonists for M1 and M3 receptors(pirenzepine and 4-DAMP, respectively)  have not able to counteract the APE effects. A sentence has been added in discussion.

Muscarine is the specific non selective agonist binding all muscarinic receptors.

Unfortunately the number of isolated Schwann cells for each patient is rather limited and for this reason we have chosen to make a first characterization of muscarinic receptors and  M2 subtype which it has been  the receptor that we have best characterized in our previous works (Loreti et al, 2006; Loreti et al, 2007; Uggenti et al, 2014; Piovesana et al, 2019). In order to carry out the reported experiments, we have developed an experimental plan using the cells derived from the same patients. For this reason we cannot currently perform additional experiments that would require the preparation of other Schwann cell cultures from other patients.

I am quite surprised that the paradoxical results obtained with APE and muscarine are only very partially discussed (L181-183) and without any reference. This part needs to be both improved and better discussed. I ask the authors to rewrite it by arguing it. Moreover, the discussion does not in any way consider explaining the results obtained: how does the specific stimulation of M2 mAChRs lead to an inhibition of cell growth? Mention should be made of the intracellular signaling induced by the activation of M2. This point is not covered in the article. The level of the discussion is poor as it is.

The discussion has been expanded.

Minor   All the minor revisions have been modified

L46: IP3 and cAMP levels

L53: have demonstrated

L55: “modulates NGF production and maturation”. Maturation of what ? Central nervous system ?

L77: As showed in Fig. 1B, M2

Reviewer 3 Report

The authors recently reported the muscarinic acetylcholine receptor M2’s function in cell growth, migration, and differentiation of rat Schwann-like adipose-derived stem cells (Page 11, line 347, reference 17). The present manuscript is a kind of follow-up study of the M2 receptor’s function in human Schwann cells. In this manuscript, the authors report isolation and cell culture of Schwann cells from human adipose obtained from three female patients undergoing reconstructive surgery with informed consent and approval by the National Research Ethics Committee, UK. They found that the human Schwann cells expressed mRNA for muscarinic acetylcholine receptor M1-5. The in vitro proliferation of the Schwann cells was inhibited by arecaidine propargyl ester (APE), a M2 receptor agonist (Figure 1). On the other hand, muscarine increased proliferation of the Schwann cells after 5 and 7-days culturing. APE’s effect on the proliferation was dose-dependent (Figure 2). Both APE and muscarine were not toxic for the Schwann cells at the concentration. Moreover, APE increased the average size of the cells after 24h of treatment (Figure 2). Furthermore, APE increased Egr-2 protein expression in the cells (Figure 3). Overall, the major M2 receptor’s function in human Schwan cells was mostly reproduced as previously reported in rat Schwann-like adipose-derived stem cells. Contrast to the previous comprehensive report, the experiments in this manuscript were only focused on analyzing the M2 receptor’s major effect. The following major and minor comments are for improving the quality of this manuscript.

Major

  1. The muscarinic acetylcholine receptor M2 and M4 are G-protein-coupled inhibitory metabotropic receptors, whereas M1, M3, and M5 are G-protein-coupled acceleratory metabotropic receptors. Although the authors found that the M2 inhibited the proliferation of the Schwann cells and increased of the cell size, they did not analyze the acceleratory metabotropic receptors’ function on the cells. Since the non-selective agonist muscarine increased the proliferation after 5 and 7-days culturing, acceleratory metabotropic receptors, such as M1, may contribute to this opposite effect. Please use M1 agonist and analyze the effect on the proliferation of the human Schwann cells.
  2. The effects of muscarinic acetylcholine receptor antagonists were not analyzed. Please analyze the effects of muscarinic acetylcholine receptor antagonists. Alternatively, discuss about the effects of muscarinic acetylcholine receptor antagonists with references.
  3. Since RT-PCR is not quantitative experiments, please quantify the M1-M5 mRNA by quantitative real-time PCR (qPCR) as shown in their previous paper. Alternatively, western blot analysis of M1-M5 receptors a good way to quantify the expression levels.

Minor

  1. The resolution of the figures is low for publication, especially Figure 3. Please revise them to high resolution figures.
  2. Page 8, line 262. The detecting method of the western blot signals is missing. Did you use an X-ray film or Cooled-CCD camera system to detect the chemiluminescence signals? Please describe it.
  3. Supplementary Material S1 appears in both text and Supplementary Material. Please solve this redundancy.

Author Response

Replies to reviewer’s comments

The authors thank all the reviewers for their comments and suggestions.

We have modified our manuscript in all parts indicated in red in the revised version, expanded discussion, modified Fig. 1 and 3, and added supplementary Fig. 1 and some new references, where required.

Rev 3

The authors recently reported the muscarinic acetylcholine receptor M2’s function in cell growth, migration, and differentiation of rat Schwann-like adipose-derived stem cells (Page 11, line 347, reference 17). The present manuscript is a kind of follow-up study of the M2 receptor’s function in human Schwann cells. In this manuscript, the authors report isolation and cell culture of Schwann cells from human adipose obtained from three female patients undergoing reconstructive surgery with informed consent and approval by the National Research Ethics Committee, UK. They found that the human Schwann cells expressed mRNA for muscarinic acetylcholine receptor M1-5. The in vitro proliferation of the Schwann cells was inhibited by arecaidine propargyl ester (APE), a M2 receptor agonist (Figure 1). On the other hand, muscarine increased proliferation of the Schwann cells after 5 and 7-days culturing. APE’s effect on the proliferation was dose-dependent (Figure 2). Both APE and muscarine were not toxic for the Schwann cells at the concentration. Moreover, APE increased the average size of the cells after 24h of treatment (Figure 2). Furthermore, APE increased Egr-2 protein expression in the cells (Figure 3). Overall, the major M2 receptor’s function in human Schwan cells was mostly reproduced as previously reported in rat Schwann-like adipose-derived stem cells. Contrast to the previous comprehensive report, the experiments in this manuscript were only focused on analyzing the M2 receptor’s major effect. The following major and minor comments are for improving the quality of this manuscript.

Thank you for the comments. Unfortunately the number of isolated Schwann cells is rather limited and for this reason we have chosen to make a first characterization of muscarinic receptors and M2 subtype, which has been the receptor that we have best characterized in our previous works (Loreti et al, 2006; Loreti et al, 2007; Uggenti et al, 2014; Piovesana et al, 2019). In order to carry out the reported experiments, we have developed an experimental plan using the cells derived from the same patients. For this reason we can not currently perform additional experiments that would require the preparation of Schwann cell cultures from other patients. For all these reasons we have chosen to proceed with a Communication that have allowed us to demonstrate for the first time the cholinoceptivity of human Schwann cells.

Major

  1. The muscarinic acetylcholine receptor M2 and M4 are G-protein-coupled inhibitory metabotropic receptors, whereas M1, M3, and M5 are G-protein-coupled acceleratory metabotropic receptors. Although the authors found that the M2 inhibited the proliferation of the Schwann cells and increased of the cell size, they did not analyze the acceleratory metabotropic receptors’ function on the cells. Since the non-selective agonist muscarine increased the proliferation after 5 and 7-days culturing, acceleratory metabotropic receptors, such as M1, may contribute to this opposite effect. Please use M1 agonist and analyze the effect on the proliferation of the human Schwann cells.

Of course the implication of M1 and M3 receptors may be responsible of the increased cell proliferation. As explained before at least we cannot perform additional experiments on the cells of these patients presented in this study. In the next work, using the cells obtained from other patients it will certainly be of our interest to evaluate the effects produced by M1 and M3 receptors using selective agonists where possible.

  1. The effects of muscarinic acetylcholine receptor antagonists were not analyzed. Please analyze the effects of muscarinic acetylcholine receptor antagonists. Alternatively, discuss about the effects of muscarinic acetylcholine receptor antagonists with references.

The ability of the APE to selectively bind M2 receptors has been previously characterized in rat Schwann cells and Schwann-like cells, using M2 antagonists gallamine and methoctramine (see Loreti et al, 2007, Uggenti et al, 2014 and Piovesana et al, 2019). Moreover we have also demonstrated that the antagonists for M1 and M3 receptors (pirenzepine and 4-DAMP, respectively) were not able to counteract the APE effects. A sentence has been added in discussion.

  1. Since RT-PCR is not quantitative experiments, please quantify the M1-M5 mRNA by quantitative real-time PCR (qPCR) as shown in their previous paper. Alternatively, western blot analysis of M1-M5 receptors a good way to quantify the expression levels.

We have performed the qRT-PCR  analysis of muscarinic receptor subtypes, but  considering the high variability of human samples appears difficult to mediate the data obtained from different patients for each single receptor subtype. For this reason we have preferred to show the single RT-PCR obtained from each patient.

However, we present the results obtained by qRT-PCR in the supplementary figure 1. The data reported confirm the variability of the different receptor subtypes as also indicated by semiquantitative RT-PCR reported in Fig. 1B. It is important to note that the data of the expression has been indicated as ΔCT, therefore the sample with the higher cycle are the receptors less expressed.

Minor

  1. The resolution of the figures is low for publication, especially Figure 3. Please revise them to high resolution figures.

Thank you for the comment. We have attached the tiff format of all figures when we have submitted the manuscript. However we have replaced the tiff in the manuscript template.

  1. Page 8, line 262. The detecting method of the western blot signals is missing. Did you use an X-ray film or Cooled-CCD camera system to detect the chemiluminescence signals? Please describe it.

We are sorry, we realized later that we did not include the description of how the acquisition of the western blot signal took place. The specifications were included in the Materials and Methods section

  1. Supplementary Material S1 appears in both text and Supplementary Material. Please solve this redundancy.

Thank you for your comment. We have enclosed the supplementary materials in the template as indicated and as supplementary file. Probably will be sufficient to remove the supplementary materials as file attached during re- submission.

Reviewer 4 Report

This manuscript provides interesting results, however there are several concerns on this manuscript as follows.

Major concerns:

1) In terms of APE 100uM treatment, the number of cells in APE 7DIV increased compared to those in APE 3 DIV (Fig. 2A). Indeed, you have described this in Line 98-101. On the other hand, the number of cells in APE 7DIV trend to decrease compared to those in APE 3 DIV (Fig. 2B). These data were obtained from any experiments under same conditions, however there is a difference. The authors should explain about this.

2) In Fig. 2B, reduced cell proliferation in APE 50uM-treatment began to show from 7DIV. Could you explain this?

Minor points:

1) In Fig. 3, you should change the high-resolution images, because we cannot confirm the data.

Author Response

Replies to reviewer’s comments

The authors thank all the reviewers for their comments and suggestions.

We have modified our manuscript in all parts indicated in red in the revised version, expanded discussion, modified Fig. 1 and 3, and added supplementary Fig. 1 and some new references, where required.

Rev. 4

This manuscript provides interesting results, however, there are several concerns on this manuscript as follows.

Major concerns:

  • In terms of APE 100uM treatment, the number of cells in APE 7DIV increased compared to those in APE 3 DIV (Fig. 2A). Indeed, you have described this in Line 98-101. On the other hand, the number of cells in APE 7DIV trend to decrease compared to those in APE 3 DIV (Fig. 2B). These data were obtained from any experiments under same conditions, however there is a difference. The authors should explain about this.

Thank you for your comments. In effect there is a difference among the two experiments reported in Fig. 2A and 2B. The complete explanation has been reported in the results section, please see line 111.

Considering this apparent increase of cell growth upon 7 days of APE 100 µM treatment, in order to evaluate if the effect may be dependent on reduced activity of M2 agonist during 7 days in vitro, we performed the same experiment at different concentrations of APE, modifying the experimental plan with the media change with or without APE treatment at the third day of culture. In this experimental condition, differently to that observed in the previous experiment reported in Fig. 1A, we observed that the cell growth resulted unmodified at 3DIV and 7DIV after 100 µM APE, confirming the inhibitory effect of the high dose of APE on cell growth.

In Fig. 2B, reduced cell proliferation in APE 50uM-treatment began to show from 7DIV. Could you explain this?

We have added a sentence in the discussion. Please see line 193.

Moreover we have observed that 50 µM APE is also able to decrease cell growth only after 7 DIV. This may be dependent on distinct reasons: 1. The affinity of the M2 receptor for APE may vary according to the glycosylation grade of the receptors in the different patients, as observed in western blot analysis; 2. The prolonged exposure to 50 µM APE may be necessary to activate the M2 downstream signalling transduction pathway.

Minor points:

  • In Fig. 3, you should change the high-resolution images, because we cannot confirm the data.

Thank you for the comment. We have attached the tiff format of all figures when we have submitted the manuscript. However we have replaced the tiff in the manuscript template.

Round 2

Reviewer 1 Report

The authors performed all required changes in this submission.

Author Response

All authors are grateful for your revision.

Reviewer 2 Report

I've read the author's comments and the revised version. I think there remains one central point on which this communication is built: the specificity of the molecule used, arecaidin propargyl ester, for M2 receptors.

L51-52 and 91-93: The authors indicate that APE is a specific agonist of M2 receptors. The 3 references indicated to argue it (10, 16, 17) are self-citations:

In doi:10.1002/dneu.22161, I read “In this study we analyzed the in vitro modulation, by the M2 agonist arecaidine”, which does not show that APE specifically binds M2 receptors.

In doi:10.1016/j.intimp.2015.05.032, it is indicated that “Recently we have demonstrated that the activation of M2 muscarinic receptors, through arecaidine propargyl ester”. Again, no data on APE/M2 binding is provided.

In doi :10.1038/s41420-019-0174-6, it is stated that “M2 receptor activation, by the preferred agonist arecaidine propargyl ester (APE), caused a reversible arrest of”. The paper does not show a specific activity of APE towards M2 receptors.

Nothing supports the fact that this molecule acts specifically on M2 receptors. Each publication seems to build on the previous one, resulting in data accepted by the authors, but obviously truncated.

On the contrary, in the publication by Jakubik et al., 1997 (Molecular Pharmacology 52: 172–179), it is shown that the APE binds with comparable affinities to the M1-4 receptors (displacement of the binding of the 3H N-methylscopolamine). This is probably what led IUPHAR (consulted on 19.08.2020) to consider that the EPA acts as a full agonist for the M1, M2, M3 and M4 receptors.

See https://www.guidetopharmacology.org/GRAC/LigandDisplayForward?tab=biology&ligandId=295

Consequently, the use of this non-specific molecule calls into question the results obtained since the cells used express the M1-M5 receptors. I suspect that the APE exerts a differential agonist effect on each of these receptors, making it difficult to draw clear conclusions about its antiproliferative effects. One could even imagine that the APE acts on receptors other than mAChRs, since the authors did not use specific antagonists of these receptors to inhibit the anti-proliferative effects. The article simply compared the effects of EPA and muscarine to conclude that these two molecules exert different effects, which has been shown in several other of their publications on non-human cells.

Author Response

please see attchment

Reviewer 3 Report

The quality of the figures was improved. The discussion and other text were revised. Although the authors did not perform the requested additional experiments, the current manuscript will fit for the communication section in this journal. At the proof step, please make sure that the supplementary file is the revised version because the current “Download supplementary files(s)” on the web is still the old version.

Author Response

We are sorry, the new supplementary materials have been added both in the template of the manuscript ad as file attached.

Round 3

Reviewer 2 Report

I read the author's comments on my second review. I have to say that I find them relatively unconvincing, as their use of pharmacological agents is supposed to be specific but can actually be questioned.

I note that, contrary to what is indicated in the rebutal letter, the article doi:10.1016/j.intimp.2015.05.032 does not contain any "competition binding experiments", but more simply experiments of "pharmacological competition", i.e cell viability, with molecules believed to act selectively. In fact, the authors note in this article about AGE “The ability of arecaidine to bind M2 receptor subtype has been previously reported”. There is therefore no evidence of binding of this agent by competitive binding assays.

I can also add that gallamine, which is qualified as "antagonists for M2 receptors" is rather a negative allosteric modulator, which also acts as antagonists of nicotinic receptors expressing the α1 subunit.

I therefore feel that the data obtained in this study are the result of pharmacological interpretations which are based on unstable assumptions. I let the editor decide the fate of this article, which presents many results, but the validity of which can be questioned.